# The Effect of Different Thermomechanical Treatments on the Metastable Phase in a Cu-Ni-Be Alloy

**DOI:** 10.3390/ma18040839

**Published:** 2025-02-14

**Authors:** Jinwen Xu, Qinde Yuan, Junbo Jia, Tianhong Wang, Yubo Shen, Zhiyuan Zhu

**Affiliations:** 1School of Materials Science and Engineering, Jiangsu University of Science and Technology, Zhenjiang 212003, China; 221110601127@stu.just.edu.cn (J.X.); 221210601223@stu.just.edu.cn (Q.Y.); 221210601309@stu.just.edu.cn (J.J.); 221210601124@stu.just.edu.cn (T.W.); 221210601411@stu.just.edu.cn (Y.S.); 2Institute of Materials, Henan Academy of Sciences, Zhengzhou 450052, China

**Keywords:** γ′, γ″, microstructure, aging process

## Abstract

This study primarily investigated the microstructural and mechanical properties of Cu-Ni-Be alloys subjected to thermomechanical treatments at 30% and 75% deformation levels. Precipitates in Cu-Ni-Be alloys are dominated by Ni-Be phases. The misfit between the Ni-Be phase/Cu interface is 0.12%. Experimental observations have revealed the existence of three classical orientation relationships between precipitates and the matrix: (110)p//(100)α; [110]p//[001]α, (110)p//(010)α; [110]p//[001]α, and (110)p//(100)α; [001]p//[001]α (p: precipitates, α: α-Cu supersaturated solid solution). Additionally, a fourth orientation relationship, (110)p//(1-1-1)α; [110]gp//[1-1-1]α (gp: Guinier–Preston), induced by deformation, has also been identified. The width of the second phase was found to be two to three atomic layers. Under 75% deformation, a substantial amount of the γ′ phase emerged at grain boundaries. Notably, at neither 30% nor 75% deformation levels were prominent cellular structures observed.

## 1. Introduction

Cu-Be alloys exhibit non-magnetic properties along with remarkable electrical conductivity, a low elastic modulus, high wear resistance, and superior mechanical properties [1]. Furthermore, Cu alloys possess high thermal conductivity, strength, ductility, and corrosion resistance under certain conditions [2,3,4]. Due to their exceptional performance, copper alloys are widely employed in industries such as the automotive, aerospace, and nuclear energy industries, as well as in sectors like electricity, telecommunications, and transportation. Prior studies have identified three orientation relationships in Cu-Ni-Be alloys: (110)p//(100)α; [110]p//[001]α, (110)p//(010)α; [110]p//[001]α, and (110)p//(100)α; [001]p//[001]α. In other alloys, deformation can lead to the emergence of multiple variants due to defect factors, a phenomenon frequently reported in Ni-based superalloys. Wei and their colleagues [5] discovered 24 variants in Cu-Be alloys, with formations closely related to deformation. Additionally, Monzen and Fang [6,7] investigated the influence of varying deformation degrees on the secondary phase. Within deformed alloys, on the one hand, defects such as dislocations facilitate the generation of variants, resulting in increased lattice distortion and enhanced macroscopic physical properties. On the other hand, these defects can open diffusion channels, accelerating the coarsening and growth of the secondary phase. For Cu-Ni-Be alloys, coarsening of the secondary phase, particularly the formation of a stable γ phase, can deteriorate an alloy’s physical properties, which is undesirable in practical engineering applications. Therefore, investigating the microstructure and properties of Cu-Ni-Be alloys subjected to thermomechanical treatments at different deformation levels is of great significance for engineering applications. Historically, determining the atomic positioning of Be and elemental concentrations in age-precipitated phases within Cu-Ni-Be alloys has been a relatively challenging task. However, a recent study utilizing integrated differential phase-contrast scanning transmission electron microscopy (iDPC-STEM) successfully addressed this challenge, and clearly observed that the Ni-Be phase is composed of alternating layers of Be and Ni atoms.

Currently, research on the microstructure of Cu-Ni-Be alloys directly aged after thermomechanical treatment is still insufficient. Existing studies have a limited understanding of the orientation relationship between the Ni-Be phase and the matrix. In this study, transmission electron microscopy (TEM) and high-resolution transmission electron microscopy (HRTEM) techniques were employed to investigate the precipitation behavior of Cu-Ni-Be alloys under different thermomechanical treatment conditions. This included the morphology, size, and four types of orientations of precipitates, as well as the misfit at the interface between the second phase and the α-Cu matrix. Additionally, the effects of precipitates on the properties of Cu-Ni-Be alloys under different deformation conditions were examined.

## 2. Materials and Methods

In this study, a Cu-Ni-Be alloy was fabricated using medium-frequency vacuum induction melting technology. The precise chemical composition of the alloy, including 2.16 wt.% Ni and 0.272 wt.% Be, was determined using an optical emission spectrometer (ZEISS, Bavaria, Germany), as detailed in Table 1. The selected specimen was a 16 mm diameter alloy rod, which was produced through a series of processes including pouring molten material within a specific temperature range (1190–1210 °C), hot forging, hot extrusion, and cold drawing. To further refine the alloy’s properties, the sample underwent a 60 min solution treatment at 980 °C, followed by hot extrusion deformation to 15% and 75% strain levels, and subsequent water quenching. The sample was then aged at 470 °C for 3 h before being naturally cooled to room temperature. For TEM observation, a 0.5 mm diameter disk was cut from the aged alloy using electrical discharge machining (EDM, LOMGKAI, Suzhou, China). This disk was gradually thinned to approximately 80–100 µm using SIC waterproof abrasive papers (abrasive paper, Suzhou, China). Ultimately, the sample was fashioned into a 3 mm diameter thin foil and further reduced to an approximate 50 µm thickness using 4000-grit abrasive paper. Electropolishing was performed in a nitric acid/methanol (1:4 volume ratio) electrolyte at 60 mA and −30 °C, followed by 30 min of ion milling at 4 kV and a 4° ion incidence angle using a Gatan 691 ion miller (Gatan, Pleasanton, CA, USA). TEM and HRTEM observations of the processed thin foil were conducted using a JEOL JEM 2100 transmission electron microscope (JEOL, Tokyo, Japan) operated at 200 kV. Bright-field (BF) images and selected area diffraction patterns (SADPs) were captured at various electron beam incidence angles to analyze the alloy’s microstructure in detail. Furthermore, fast Fourier transform (FFT) and inverse FFT (IFFT) techniques in Digital Micrograph software (version 3.4, Gatan, Pleasanton, CA, USA) were employed for additional image analysis of HRTEM images. In addition to the microstructural analysis, the alloy’s metallographic microstructure was observed using an optical microscope (OM, ZEISS, Oberkochen, Germany), and its hardness was measured using an automatic hardness tester (KB 30S, KB Prüftechnik, Hochdorf-Assenheim, Germany). The alloy’s electrical conductivity was also tested using an FD-101 digital portable eddy current conductivity meter (FD-101, Xingsha, Xiamen, China), providing a comprehensive assessment of its physical and mechanical properties.

## 3. The Effect of Different Amounts of Thermal Deformation on Structure and Properties

In traditional processes, alloys are cast, forged, deformed, solution treated, and aged for use. However, in this experiment, the cast metal was directly solution treated, kept warm at 950 °C for 1 h, and then hot extrusion was carried out immediately at 950 °C, deformation was controlled between 15~75%, and the strain rate was 1 s^−1^. Optical micrographs of Cu-Ni-Be alloy in solution-aged condition are presented in Figure 1. The aging treatment was conducted at 470 °C for 3 h. For the undeformed alloy, the grain size distribution ranged from 80 to 200 µm (Figure 1a). In contrast, the alloy subjected to 30% deformation exhibited a grain size between 10 and 75 µm (Figure 1b), while the 75% deformed alloy displayed a grain size range of 5 to 50 µm (Figure 1c). All samples featured fine equiaxed grains resulting from recrystallization. Notably, no significant nodular structures, also known as grain boundary reactions characterized by discontinuous precipitates of the Ni-Be phase, were observed at these grains’ boundaries. At the same thermal deformation temperature, varying degrees of deformation significantly impact Cu-Ni-Be alloys. As the amount of hot deformation increases, the grain size refines, accompanied by the formation of annealing twins and mechanical twins during the thermal deformation process. The introduction of deformation leads to dislocation pile-ups at grain boundaries, forming substructures such as dislocation walls or networks. These defects contribute to the enhanced electrical conductivity of the material. In the study by Monzen [4], it was found that the hardness of cold-deformed Cu-Ni alloys with low Be content is consistently higher than that of undeformed samples. For Cu-Ni-Be alloys, post-deformation aging results in a rapid increase in conductivity due to the increased continuous precipitation of phases. During the solid solution process, Ni alloying elements dissolve into the Cu matrix, forming an oversaturated solid solution α-Cu. The incorporation of solute atoms into the solid solution can cause lattice distortions and misfit dislocations. Figure 2 shows the graph of hardness and electrical conductivity of a Cu-Ni-Be alloy after time effecting at different heat deformation levels. These experimental results show that the hardness and electrical conductivity of the alloy increased continuously as the deformation level ranged from 15% to 30%. At a deformation level of 30%, the hardness and electrical conductivity reached their peak values (respectively, 276 HV5 and 56.4 IACS). Grain refinement is the main factor that leads to an increase in hardness after heat deformation. The hot extrusion process, in addition to eliminating pores and micro-crack defects in the cast-state alloy to improve density, can also make the grains finer and eliminate a certain degree of macrosegregation. For Cu-Ni-Be alloys with deformation levels of 45% to 75%, hardness continued to decrease. This may be related to their microstructure or dislocation slip ability, and excessive deformation may have destroyed their excellent organization. This may be the reason why the physical properties of the alloy decreased after the 45% deformation level. Similar to the hardness curve, the electrical conductivity of the material reached its peak at 30% deformation due to the increased density, and then decreased. Unexpectedly, the electrical conductivity showed an upward trend after 45% deformation.

In Equation (1), $d_{\alpha }$ and $d_{\beta }$ represent the inter-planar distances of the precipitated phase and the matrix, respectively; and *δ* denotes the interfacial mismatch value between the two phases. The bright-field image (a) of the 30% deformed sample in Figure 3 reveals the deformation of the precipitated phase, with a disc-like morphology changing to a curved filament and spherical shape. Figure 3b shows d-spacings d(002)α and d(010)p, which were calculated using microdensitometry trace lines [8] d(002)α = 1.31 nm and d(010)p = 1.16 nm. The degree of mismatch was then determined using the mismatch formula. The degree of mismatch between the precipitated phase Ni-Be and the Cu matrix was 0.12%. At that time, in the direction of the long axis, the interface between the matrix and the second phase was a semi-coherent interface(1)$$\delta =\frac{2\left(d_{\alpha }-d_{\beta }\right)}{\left(d_{\alpha }+d_{\beta }\right)}$$

The precipitation sequence of Cu-Ni-Be alloys is as follows: α-Cu supersaturated solid solution → (gp) zones → γ″ → γ′ → stable γ (Ni-Be) phase. The GP zones in Cu-Ni-Be alloys consist of a single layer of disk-shaped Be atoms parallel to the {100}αCu matrix. The aging precipitate phase γ″ is alternately formed by Be and Ni atoms, without undergoing a change in habit plane. In previous studies by Zhu Zhiyuan, Cai Yuanfei, et al., the γ″ phase precipitated during aging and exhibited three orientation relationships with the matrix, which can be represented as Variant A: (110)p//(100)α; [110]p//[001]α, Variant B: (110)p//(010)α; [110]p//[001]α, and Variant C: (110)p//(100)α; [001]p//[001]α [9]. This experiment conducted high-resolution characterization on a 30% deformed sample. Combined with panel (a) of Figure 4, FFT was used to locate GP zones, and a γ″ phase consisting of two layers of Be atoms, as shown in Figure 4d, was clearly observed in the GP zone at the top right corner. Its thickness was measured to be 0.411 nm, as shown in Figure 4e. Its long axis was parallel to the (1-1-1)α of the matrix, and its orientation relationship with the matrix can be expressed as: (110)p//(1-1-1)α; [110]gp//[1-1-1]α. Similarly, as mentioned in [10], it is believed that the GP zone first forms a γ″ approximately 10 nm long, consisting of two rows of Be atoms. It is also noteworthy that a GP zone composed of Be atoms appeared in the upper right corner of the GP zone in region 1, with a transition zone of approximately 4 nm. Figure 4f shows that the (1-11)α of the matrix underwent a certain degree of deformation. Figure 4b presents the morphology of GP zones precipitated along the {111} slip planes in different regions. One type is the γ″ phase arranged by two Be atoms within the region (region 1) or a GP zone arranged by a single layer of Be atoms (above region 1), and the other is a short clustered lamellar structure (region 2), both maintaining a fully coherent form with the matrix. In Figure 4e, it can be clearly observed that within the GP zone the overall lattice spacing was relatively uniform, around 0.215 nm. After passing through the γ″ phase composed of two Be atoms, the peak intensity significantly increased, and the overall thickness was 0.411 nm. Figure 4f shows the lattice fringe image of another GP zone morphology, where the lattice spacing was not uniform, but ranged from 0.182 to 0.251 nm, indicating a substantial coherent stress field at this stage. In the bright-field image of a 75% hot-deformed sample, as shown in Figure 5, the second phase grows along the dislocation lines in a curved manner. By analyzing the diffraction patterns in Figure 5b, it was determined that the long axis of the second phase was parallel to the (100)α, (010)α, and (110)α directions of the matrix. In the high-resolution images of Figure 5c–e, the classic three variants can be observed. However, at the end of the second phase in Figure 5d, its precipitation direction is parallel to [110]α. Unlike previous experiments, under a deformation of 75%, second phases with lengths exceeding 100 nm existed, as shown in Figure 5f,g. These were confirmed to be the γ′ phase through high-resolution transmission electron microscopy images and FFT patterns, as shown in Figure 5h–j.

## 4. Analysis and Discussion

In this experiment, no significant cellular structures were observed. This is favorable, as it indicates that precipitates did not undergo excessive coarsening to develop into the γ phase. The emergence of such structures would significantly degrade the overall mechanical properties of the material. Many studies [7,8,9,10,11] have suggested that the Cu-Ni-Be alloy exhibits four variants in the orientation relationship between precipitates and the matrix: (110)p//(100)α; [110]p//[001]α, (110)p//(010)α; [110]p//[001]α, (110)p//(100)α; [001]p//[001]α, and (110)p//(1-1-1)α; [110]p//[1-1-1]α. The emergence of the fourth variant is attributed to deformation. The material in this experiment was obtained by direct aging after homogenization treatment. The appearance of the fourth orientation relationship could hinder dislocation motion, increasing the material’s hardness, but reducing its plasticity. As the close-packed plane of Cu alloys is {111}, the high vacancy concentration on the {111}α slip plane during the early stages of aging facilitated the segregation of solute atoms to form disk-shaped GP zones. The coexistence of γ″ phases precipitating from both {111} and {100} planes created a high coherent stress field, leading to the highest strengthening effect in the Cu matrix, which manifested as peak macroscopic hardness at 35% deformation.

In the study by Phillips et al. [12], the primary reason for the decrease in electrical conductivity in Cu-Ni-Be samples was attributed to the formation and expansion of GP zones, which enhanced electron wave scattering. The segregation of solute atoms and the presence of coherent relationships intensified this scattering effect. In our experiments, the electrical conductivity reached its minimum at 45% deformation, suggesting that this is the process regime where GP zones fully develop. Dislocations introduced by deformation may cause GP zones to be sheared or dissolved by dislocations, releasing solute atoms, which is equivalent to a re-solid solution process. If these dislocations do not completely destroy the GP zones, the interface between the dislocations and the GP zones form an electron scattering barrier. The increase in electrical conductivity at 60% and 75% deformation was attributed to the dynamic recovery process, the coarsening of GP zones, the transformation of precipitates, and texture anisotropy. During the dynamic recovery process, dislocations rearrange into low-energy structures (such as dislocation walls and subgrain boundaries) through climb and cross-slip, significantly reducing the effective scattering density. High strain promotes the coarsening of GP zones or accelerates the transformation of precipitates, reducing the interface between precipitates and the matrix (decreasing the interface area per unit volume), thereby weakening electron scattering. Dislocations introduced by deformation provide more nucleation sites for GP zones, resulting in a higher density of precipitates and a decrease in conductivity. The subsequent increase in conductivity at 60% and 75% deformation was attributed to the presence of γ′ phases in these samples. In our experiments, γ′ phases with long axes ranging from 100 to 200 nm were observed in bright-field images of 75% deformed samples. This is consistent with the findings of Monzen, Yu, and Peng [6,13,14] that stress application facilitates the opening of diffusion channels, accelerating the nucleation and coarsening of γ″ and γ′ phases. The authors attribute this phenomenon to the diffusion rate at the interface between the matrix and precipitates. The process of γ″ agglomeration and growth into γ′ phases leads to a decrease in the density of precipitates, thereby increasing the electrical conductivity of the samples. Finally, texture plays a role; deformation-induced strong texture (copper texture) results in a preferred orientation of the current direction on the {112} plane, reducing anisotropic resistance.

For Cu-Be alloys, the orientation relationship between precipitates and the matrix can transform from a Bain to an NW relationship during aging. However, such a transformation does not occur in the Ni-Be phase of Cu-Ni-Be alloys [10,12,13,14]. Due to matrix deformation, the second phase precipitates and grows along dislocation lines. In this study, the misfit between the Ni-Be phase and the Cu matrix was 0.12%. It belongs to a coherent interface, which means there was lower interfacial energy and a higher interfacial migration rate. The lower interfacial energy was conducive to nucleation, while the higher interfacial migration rate enabled it to grow rapidly into an elliptical shape. In terms of stability, the Ni-Be phase exhibited high thermal and mechanical stability, with a slow coarsening rate.

## 5. Conclusions

The aging precipitation process of the sample Cu-Ni-Be alloy followed the sequence of gp zone → metastable γ″ → γ′ → stable γ. The hardness value first increased and then stabilized. The electrical conductivity initially rose, reached a peak at 30% deformation, subsequently decreased to a minimum at 45% deformation, and then rose again. Specifically, the hardness of the sample reached its peak when deformation reached 30%.

The fourth orientation relationship between precipitates and the matrix arose from the thermomechanical treatment process, leading to multiple orientation relationships that maximized macroscopic hardness.At 45% deformation, the full development of GP zones in the alloy reduced electrical conductivity. When deformation reached 75%, the presence of γ′ phases with coherent interfaces with the matrix enhanced electrical conductivity while reducing hardness.In this alloy, the second phase grew along dislocation lines, and increased deformation accelerated the coarsening process of γ″ phases.

## Figures and Tables

**Figure 1 materials-18-00839-f001:**
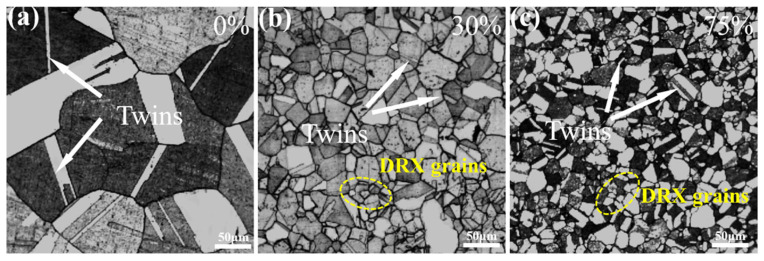
Microstructures of the Cu-Ni-Be alloy after aging under the same conditions following different thermal deformations: (**a**) undeformed, (**b**) 30% deformation, and (**c**) 75% deformation.

**Figure 2 materials-18-00839-f002:**
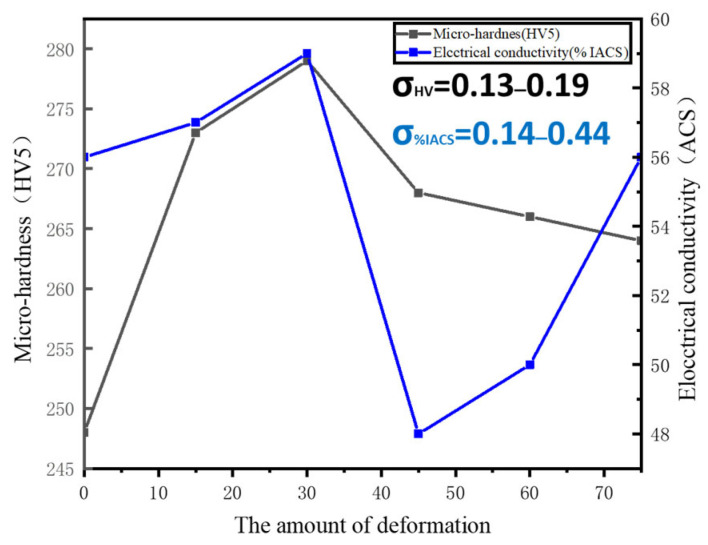
Hardness and electrical conductivity of Cu-Ni-Be alloy after aging following different thermal deformations.

**Figure 3 materials-18-00839-f003:**
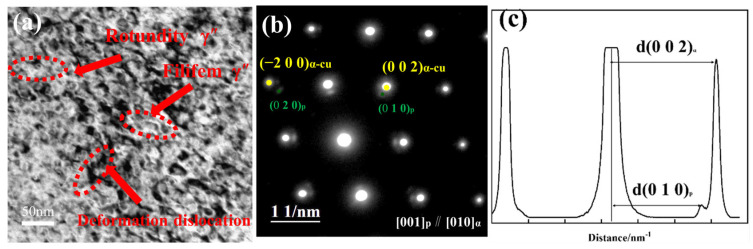
(**a**) Bright-field image of the sample with 30% deformation. (**b**) Selected area electron diffraction pattern. (**c**) Microdensitometer trace across (010)p and (002)α along the [010]α zone axis.

**Figure 4 materials-18-00839-f004:**
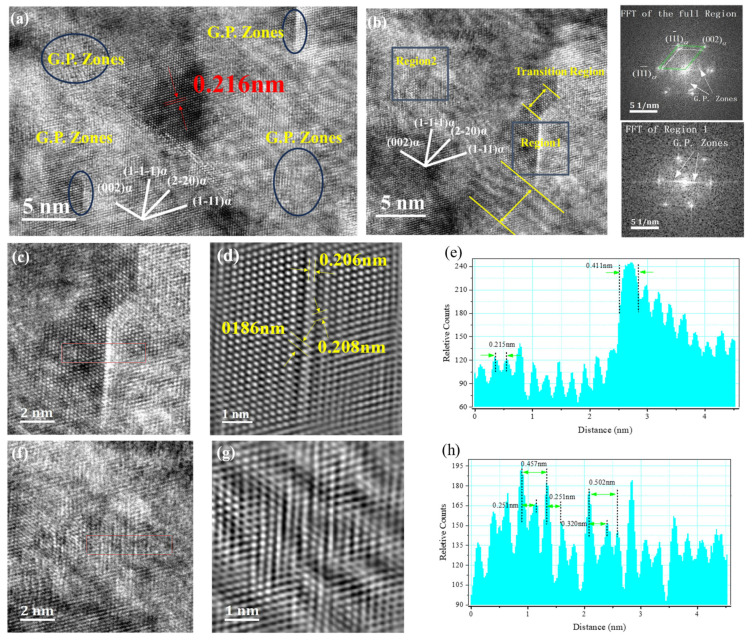
(**a**) High-resolution images of Cu-Ni-Be alloy under 75% deformation. (**b**) An enlarged view of the selected area in (**a**). (**c**,**d**,**f**,**g**) Enlarged views of selected areas in (**b**). (**e**,**h**) The corresponding lattice spacings for (**d)** and (**g**) respectively.

**Figure 5 materials-18-00839-f005:**
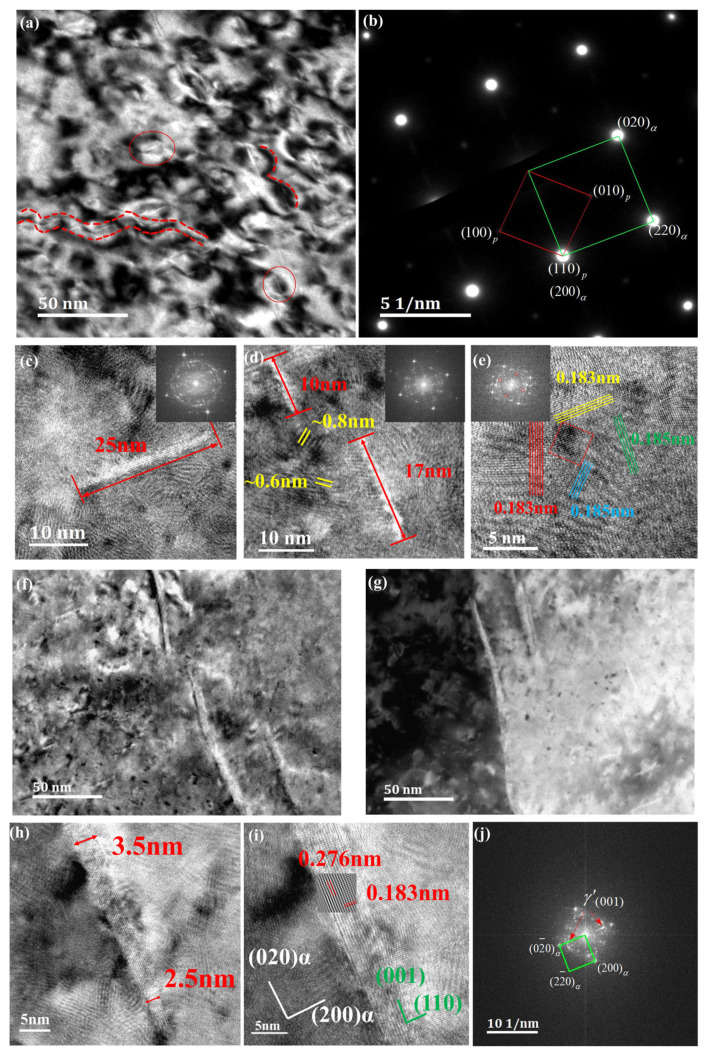
Single-atom images under 75% deformation (**a**), with the overall selected area electron diffraction shown in (**b**). High-resolution images of this selected area are presented in (**c**–**e**). Single-atom images under 5% deformation are shown in (**f**,**g**). High-resolution images of the grain boundary in (**g**) are displayed in (**h**,**i**). The FFT image of (**i**) is presented in (**j**).

**Table 1 materials-18-00839-t001:** Chemical composition of alloys used.

Element	Cu	Ni	Be	Impurty
(wt.%)	Bal.	2.16	0.272	≤ 0.1

## Data Availability

The original contribution presented in this study is included in the article, and further questions can be addressed to the appropriate authors.

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
