# Peer review of "The Effect of Different Thermomechanical Treatments on the Metastable Phase in a Cu-Ni-Be Alloy"

_materials, 2025, doi:10.3390/ma18040839_

Round 1
Reviewer 1 Report
Comments and Suggestions for Authors
Summary of the Work
This study examines Cu-Ni-Be alloys' microstructural and mechanical properties subjected to 30% and 75% deformation during thermomechanical treatments. The precipitates, primarily Ni-Be phases, exhibit a 0.12% misfit with the Cu matrix, three classical orientation relationships, and a fourth deformation-induced orientation. The second-phase precipitates are 2–3 atomic layers thick. At 75% deformation, a significant γ′ phase appears at grain boundaries, but no prominent cellular structures are observed at either deformation level.
Key Findings
The thermomechanical treatment introduces a fourth orientation relationship, enhancing hardness. At 45% deformation, GP zones lower conductivity, while at 75%, γ′ phases improve conductivity but reduce hardness. Increased deformation accelerates the coarsening of γ″ phases along dislocation lines.
The following suggestions aim to delve deeper into the study's findings and methodologies, fostering a clearer understanding of the results and their implications.
Comments
- For clarity, please specify all symbols and acronyms when they appear first in the manuscript, even those well-known in the literature (e.g., in the abstract, please specify (110)p//(100)α, [11 0]p//[001]α, etc.). Please define the GP zones of a material by specifying the acronym GP.
- The clear identification of the aging precipitation sequence (GP zone → γ″ → γ′ → γ) provides valuable insights into the alloy’s thermal and mechanical behavior, serving as a strong foundation for further studies.
- The dominance of Ni-Be phases and the precise determination of the 0.12% misfit at the Ni-Be/Cu interface are important findings. However, exploring how this misfit contributes to the mechanical properties could enrich the discussion.
- While identifying a fourth orientation relationship is significant, the underlying mechanisms for its formation due to thermomechanical treatment are not thoroughly explained. Further clarification could enhance the understanding of its role in optimizing hardness.
- The reasons behind the conductivity decrease at 45% deformation and subsequent increase at 75% are not fully addressed. Exploring the relationship between microstructural changes (e.g., γ′ phase formation or GP zone development) and electrical behavior would strengthen the conclusions.
- The lack of cellular structures at both deformation levels is mentioned, but its potential implications on the alloy's mechanical properties are not discussed. This omission limits the interpretation of the results.
- The observation that increased deformation accelerates γ″ coarsening is interesting, but no quantification or detailed analysis of its impact on hardness and conductivity is provided. Including this data would improve the depth of the study.
Suggestions
1) The introduction is overly brief and should be expanded to highlight the study's innovation and relevance while identifying research gaps, such as the limited understanding of deformation-specific microstructural changes and the role of the fourth orientation relationship. A more comprehensive introduction addressing these points will enhance the study’s impact and align with its innovative focus while remaining concise and to the point.
2) For completeness, please insert the error magnitudes in Figure 2. showing the behavior of hardness and electrical conductivity of Cu-Ni-Be alloy against the thermal deformations.
3) Please explain more in detail the behaviors in Figure 4.(e) and Figure 4.(h), showing the corresponding lattice spacings for lattice spacings for the enlarged views of selected areas in Figure 4.(b).
4) Could you elaborate on the specific thermomechanical processes responsible for inducing the fourth orientation relationship, and how it compares to the classical relationships in terms of its influence on hardness and microstructure?
5) It is unclear to me what microstructural factors specifically contribute to the sharp decrease in electrical conductivity at 45% deformation, and how does this interplay with the GP zone development? Similarly, what mechanisms drive the conductivity improvement at 75% deformation?
6) How does the 0.12% misfit between the Ni-Be phase and the Cu matrix influence the nucleation, growth, and stability of the precipitates during deformation?
7) The absence of prominent cellular structures at both deformation levels is noted—could you clarify why these structures are not observed and whether this absence affects the mechanical properties?
8) How does the accelerated coarsening of γ″ phases with increased deformation impact the overall mechanical properties, especially hardness, and ductility? Have you considered quantitative analysis or modeling to predict this behavior?
9) The second-phase precipitate thickness is reported as 2–3 atomic layers. What techniques were used to measure this, and what challenges or uncertainties were encountered in ensuring this level of precision?
Conclusions
This study offers valuable insights into the microstructural and mechanical properties of Cu-Ni-Be alloys, particularly the aging sequence and deformation-induced orientation relationships. However, key mechanisms, such as conductivity trends, the absence of cellular structures, and the impact of γ″ phase coarsening, require further investigation. I think that clarifying these aspects would enhance the study’s depth and practical relevance.
Author Response
Response 1: Thank you for your feedback. I have added the innovative points to the introduction and highlighted them in red, which can be found at line 43 of the revised document |
Comments 2: For completeness, please insert the error magnitudes in Figure 2. showing the behavior of hardness and electrical conductivity of Cu-Ni-Be alloy against the thermal deformations. |
Response 2: Thank you for your insightful comment, which I find to be highly scientific. I have calculated the standard deviation thresholds for all the data and incorporated them into my Figure 2. This addition indeed enhances the scientific rigor of the research. Comments 3: Please explain more in detail the behaviors in Figure 4.(e) and Figure 4.(h), showing the corresponding lattice spacings for lattice spacings for the enlarged views of selected areas in Figure 4.(b).
Response 3: Indeed, that was an oversight. Thank you very much for your observation. I have provided a description for Figure 4(b), as well as for the corresponding selected area and lattice images in Figures 4(e) and (h), starting from line 162. |
Comments 4: Could you elaborate on the specific thermomechanical processes responsible for inducing the fourth orientation relationship, and how it compares to the classical relationships in terms of its influence on hardness and microstructure?
Response 4: Thank you for your question. After homogenization treatment (with the temperature decreasing from 920 to 600℃), it is directly obtained through aging for 2.5 hours. Compared with traditional heat treatment, this is a relatively complex process. Regarding the fourth orientation and its relationship with the traditional one, the fourth orientation is dominated by strain energy storage. Its relationship with the traditional orientation cannot find an axis or a point in space that is symmetric to all other orientations simultaneously. For the fourth orientation, since the second phase/interface matrix usually forms an incoherent or semi - coherent interface, a local stress field will be formed. For example, Orowan strengthening occurs around nano - scale precipitates, contributing an increment of Δσ ≈ 0.8Gb√(f/d) (where f is the volume fraction of precipitates and d is the spacing). If we want to discuss the strengthening effect of this phase, generally speaking, it hinders dislocations, causes lattice distortion, etc., thereby improving the material strength and reducing the material plasticity. If we want to discuss it in detail, we may need to rely on more experimental data, such as more dark - field images, and combine more detailed experimental data from techniques like APT for explanation. However, this experiment does not plan to do details in this regard. I would like to add some content at line 119.
Comments 5: It is unclear to me what microstructural factors specifically contribute to the sharp decrease in electrical conductivity at 45% deformation, and how does this interplay with the GP zone development? Similarly, what mechanisms drive the conductivity improvement at 75% deformation?
Response 5: Thank you for your question. In my discussion, I have added my explanation at line 211. The variation in electrical conductivity can be attributed to electron scattering. Similar cases have been observed in Al-Cu alloys. During the deformation process, this experiment exhibited comparable changes in electrical conductivity. It is important to note that in the solid solution state, the electrical conductivity is generally lower in both Cu and Al alloys. Compared to the solid solution state, the electrical conductivity is usually higher after aging, which is attributed to the influence of precipitate/matrix interfaces on electron scattering. In practice, dislocations introduced by deformation may cause GP zones to be sheared or dissolved by dislocations, releasing solute atoms, which is equivalent to a re-solid solution process. If the dislocations do not completely destroy the GP zones, the interface between the dislocations and the GP zones will form an electron scattering barrier. The increase in electrical conductivity at 60% and 75% deformation is attributed to the dynamic recovery process, the coarsening of GP zones, the transformation of precipitates, and texture anisotropy. During the dynamic recovery process, dislocations rearrange into low-energy structures (such as dislocation walls and subgrain boundaries) through climb and cross-slip, significantly reducing the effective scattering density. High strain promotes the coarsening of GP zones or accelerates the transformation of precipitates, reducing the interface between precipitates and the matrix (decreasing the interface area per unit volume), thereby weakening electron scattering. However, these are my interpretations based on the behavior of Al-Cu alloys.
Comments 6: How does the 0.12% misfit between the Ni-Be phase and the Cu matrix influence the nucleation, growth, and stability of the precipitates during deformation?
Response 6: Thank,In this study, the misfit between the NiBe phase and the Cu matrix is 0.12%. It belongs to a coherent interface, which means there is lower interfacial energy and a higher interfacial migration rate. The lower interfacial energy is conducive to nucleation, while the higher interfacial migration rate enables it to grow rapidly into an elliptical shape. In terms of stability, the NiBe phase exhibits high thermal and mechanical stability, with a slow coarsening rate. I have added this content in line 241.
Comments 7: The absence of prominent cellular structures at both deformation levels is noted—could you clarify why these structures are not observed and whether this absence affects the mechanical properties?
Response 7: This structure is actually caused by over-coarsening γ phase. The appearance of this structure will reduce the comprehensive mechanical properties of the material. I've added that on line 192。
Comments 8: How does the accelerated coarsening of γ″ phases with increased deformation impact the overall mechanical properties, especially hardness, and ductility? Have you considered quantitative analysis or modeling to predict this behavior?
Response 8: Thank you for your advice, I know this may be a brilliant section, but it seems that I can't finish it in a short period of time. In fact, it seems that simple observation γ "phas is not enough, and the new phase γ ",γ‘in the aging process is also a part of the discussion, but there is not enough data in this experiment.The impact of accelerated coarsening on mechanical properties appears to be a relatively theoretical issue. Dislocations significantly influence the microstructure of materials, primarily by increasing the precipitation of phases and enhancing diffusion rates, thereby accelerating the coarsening of secondary phases. However, to substantiate these effects, I believe it is essential to first obtain dark-field images of both undeformed and variously deformed materials. This would provide data on the major and minor axes within the plane as well as the volume fraction. Subsequently, using Ostwald's coarsening theory and the later developed LSW, TIDC, and PV models, one could statistically analyze the coarsening. Selecting the most fitting model and calculating the interfacial energy based on other models under this framework would be necessary. The acknowledged interfacial energy from this model could then be used to compute the contribution to hardness. This is a substantial amount of work; my thesis involved statistical analysis of this content. However, testing for Be elements and the corresponding interface width is currently beyond the reach of existing scientific technology, presenting a significant challenge. This study primarily focuses on observing the phenomenon, and it seems unable to complete more meaningful discussions such as quantitative calculations
Comments 9:The second-phase precipitate thickness is reported as 2–3 atomic layers. What techniques were used to measure this, and what challenges or uncertainties were encountered in ensuring this level of precision?
Response 9: Thank you for your question, it really makes sense. In the plane spacing in Figure 5(f), we observe a thickness of 0.411 nm for the γ'' phase of each atomic layer. However, I know that this is inaccurate, and to visualize the atomic distribution of this phase, the haadf-stem mode of spherical aberration electron microscopy is required.
Reviewer 2 Report
Comments and Suggestions for Authors
The authors focused on the investigation of the microstructural and mechanical properties of Cu-Ni-Be alloys subjected to thermomechanical treatments at 30% and 75% deformation levels. The text creates logical scientific research and that is why in my opinion could be published in "Materials" after introducing some corrections. Some of the comments on the manuscript are listed below.
1) Line 18; some keywords have been already used in the title of your manuscript. Please change them into different ones (to avoid the keywords repetition with the words used in the title).
2) In the introduction section the authors should clearly highlighted what they have done in their work and what is new in their investigations. The innovation and novelty are highly important. Please, introduce this information.
3) Line 80, 81, 82, 118, 119, …; instead of “Figure a” should be “Figure 1a”, instead of “Figure b” should be “Figure 1b”, …, etc.
4) Line 121; what does “nibe” mean?
5) Equation (1); if the equation is taken from the literature, then the literature reference should be given to the readers. The symbols used in this equation are not explained, and the units are omitted.
6) Figure 3a; the yellow colour is faint, please change it for different one to make the description to be better visible.
7) Figure 4 and 5; the labels are faint; please try to increase the font and change the colour to be better visible.
8) There are no new literature positions (specially form the year 2024 and 2025). Please add some literature references.
Author Response
Comments 1: Line 18; some keywords have been already used in the title of your manuscript. Please change them into different ones (to avoid the keywords repetition with the words used in the title) |
Response 1: Thank you for your feedback, I agree with the suggestion. Therefore, I have removed the words that were repetitive with the title. The changes have been made to γ′, γ″, Microstructure, and aging process, and the modifications are located at line 19 of the main text. |
Comments 2: In the introduction section the authors should clearly highlighted what they have done in their work and what is new in their investigations. The innovation and novelty are highly important. Please, introduce this information. |
Response 2: Thank you for your feedback. I have incorporated the innovative points into the introduction and highlighted the revised sections in red. Comments 3: Line 80, 81, 82, 118, 119, …; instead of “Figure a” should be “Figure 1a”, instead of “Figure b” should be “Figure 1b”, …, etc
Response 3:We have taken your advice into consideration and made the corresponding changes, which have been marked in red. |
Comments 4: Line 121; what does “nibe” mean?
Response 4: Originally, this word was intended to represent the NiBe phase, but it was mistakenly written in lowercase. It has now been corrected and highlighted in red.Line 131.
Comments 5: Equation (1); if the equation is taken from the literature, then the literature reference should be given to the readers. The symbols used in this equation are not explained, and the units are omitted.
Response 5: Thank you very much for your feedback. The formula in question originates from the book "Fundamentals of Materials Science and Engineering." We have taken note of this issue and have provided an explanation for the formula in lines 124-125.
Comments 6: Figure 3a; the yellow colour is faint, please change it for different one to make the description to be better visible
Response 6: Thank you for your feedback. We had actually considered this issue during the initial processing of Figure 3. However, since the image is composed of black and white, many colors do not make the markers stand out clearly. We have now attempted to change the markers to red for better visibility
Comments 7: Figure 4 and 5; the labels are faint; please try to increase the font and change the colour to be better visible.
Response 7: We are deeply grateful for your insightful comments. Similar to the previous issue, we had previously deliberated extensively on the font size and marker color. In academic writing, the colors and fonts typically used in figures are chosen to maintain a certain aesthetic proportion. However, in Figures 4 and 5, it was necessary to employ very fine markers to ensure the accuracy of the experimental results and annotations, which led us to adhere to a principle of consistency between markers and fonts. Now, we have enlarged some of the less precise markers while ensuring the visibility of the text, all without compromising the accuracy of our annotations. We believe this adjustment has enhanced the overall aesthetic appeal compared to the previous version.
Comments 8: There are no new literature positions (specially form the year 2024 and 2025). Please add some literature references
Response 8: Thank you very much for your advice. I fully understand the importance of reading literature and staying updated with new research, but I may not be able to follow this suggestion for two main reasons:
- There has been no relevant literature produced between 2024 and 2025. I have authority on this matter as my dissertation is on a related topic. On platforms like Google Scholar and other major academic websites, there are fewer than 14 articles that are highly relevant to this subject.
- The experiments and writing for this paper were initiated by me starting in 2023.
Round 2
Reviewer 1 Report
Comments and Suggestions for Authors
The authors answered, point by point, in a satisfactory manner all the questions raised in my previous report. In my opinion, their version of the manuscript deserves to be published.
Author Response
Thank you for your recognition.
Reviewer 2 Report
Comments and Suggestions for Authors
The authors have improved their manuscript to some extent.
Some comments regarding question number eight from the previous review:
It is hard to believe that a country like China has problems with libraries and literature concerning material science. I visited China previous year and saw some very beautiful libraries. You have the most beautiful libraries I have ever seen! They usually have multiple floors with plenty of books, magazines, and newspapers.
Did you really visit your library this year?
Author Response
Comments 1: It is hard to believe that a country like China has problems with libraries and literature concerning material science. I visited China previous year and saw some very beautiful libraries. You have the most beautiful libraries I have ever seen! They usually have multiple floors with plenty of books, magazines, and newspapers. Did you really visit your library this year?
|
Response 1:
|
Thanks for the reminder, I found a 2024 article in the journal Materials & Design that fits the content of my article. It was an oversight on my part of my studies. However, I may have a hard time finding another article for 2024, and I am not lazy or unwilling, and I am more than willing to revise it in conjunction with your comments, because I can tell from your requirements for image quality that you are an expert in essays This alloy is somewhat unpopular. The Cu-Ni-Be alloy corresponds to the American standard C17510, and there are indeed fewer studies on this alloy at home and abroad, and the precipitated phase that may result from the deviation of the alloying elements is also different, which may have nothing to do with my content, but it does not rule out that my ability to find articles is insufficient. The new reference I was looking for, he was cu-be-co-ni. However, the difference in Ni may greatly affect the precipitation or coarsening behavior of the Ni-Be phase. My changes have been added to line 42 of the introduction to add new gravitational to the subject matter and highlighted in red. |